# Challenges of a Patient with Thromboembolism

**DOI:** 10.3390/reports6030039

**Published:** 2023-08-22

**Authors:** Andra Oancea, Alexandra Maștaleru, Irina Mihaela Abdulan, Alexandru Dan Costache, Mădălina Ioana Zota, Robert Negru, Ștefana Moisă, Laura Mihaela Trandafir, Maria Magdalena Leon

**Affiliations:** 1Department of Medical Specialties I, “Grigore T. Popa” University of Medicine and Pharmacy, 700115 Iasi, Romania; andra.radulescu@yahoo.com (A.O.); adcostache@yahoo.com (A.D.C.); madalina.chiorescu@gmail.com (M.I.Z.); robert.negru@gmail.com (R.N.); leon_mariamagdalena@yahoo.com (M.M.L.); 2Clinical Rehabilitation Hospital, 700661 Iasi, Romania; 3Department of Mother and Child, “Grigore T. Popa” University of Medicine and Pharmacy, 700115 Iasi, Romania; stefana-maria.moisa@umfiasi.ro (Ș.M.); trandafirlaura@yahoo.com (L.M.T.)

**Keywords:** thrombophilia, factor V Leiden, venous thromboembolism, acute coronary syndrome, cardiac rehabilitation, apixaban

## Abstract

Background: FV Leiden is an autosomal dominant disease, representing one of the most prevalent genetic causes for hereditary thrombophilia manifested by venous thromboembolism. Methods: We report a case of a 30-year-old patient who was admitted for enrollment in phase II cardiac rehabilitation. The cardiovascular disease onset was five years ago when the patient was diagnosed with superficial vein thrombosis, for which anticoagulant treatment was recommended. However, he discontinued the prescribed treatment independently, which resulted in the development of deep vein thrombosis. A screening for risk factors associated with venous thromboembolism was conducted, leading to the identification of a heterozygous mutation of factor V Leiden. Later, the patient was hospitalized for acute coronary syndrome necessitating stent implantation. Following this procedure, the patient started a cardiac rehabilitation program, where the patient received multidisciplinary counseling. Conclusions: At the end of the cardiac rehab, significant improvements were observed in clinical and hemodynamic parameters. Consequently, the patient was advised to continue rehabilitation treatment in the outpatient setting. Also, for patients with suboptimal maintenance of the therapeutic range of INR, the use of apixaban might be considered. Furthermore, the utilization of a reduced dosage of apixaban has demonstrated its effectiveness in preventing further venous thromboembolism.

## 1. Introduction

Factor (F) V is a glycoprotein required for the production of the prothrombinase complex, being necessary for the clot’s development. Factor V increases the conversion of prothrombin to thrombin and, subsequently, enhances fibrin formation [1]. The protein C pathway is one of the counter-regulatory systems. Activated protein C (APC), together with its cofactor, protein S, inactivates the activated FV, inhibiting coagulation. A single-point mutation in the FV gene at position 1691G/A causes the creation of an aberrant protein (FV Leiden), leading to a prothrombotic condition due to the enhanced procoagulant role of the activated factor V and the reduced anticoagulant role of factor V [2].

FV Leiden is an autosomal dominant disease, being one of the most common causes of hereditary thrombophilia. The patients diagnosed with thrombophilia can have a heterozygous or homozygous genotype, the last being more prone to developing thrombosis. Thus, the patients that are heterozygous for FV Leiden have a five- to seven-times-higher risk of developing venous thromboembolism, while the risk increases by 25–50 times in individuals with the homozygous genotype [3]. 

The most prevalent clinical manifestation of a patient that has heterozygous factor V Leiden mutation is venous thromboembolism (some of the most frequent vascular beds affected are deep vein thrombosis [4], pulmonary embolism [5,6], superficial vein thrombosis [7], and the cerebral [8] or portal vein [9]). Moreover, studies have demonstrated correlations between factor V Leiden and arterial thromboembolism, such as myocardial infarction [10], stroke, or transient ischemic attack [11].

Implementing cardiac rehabilitation following acute coronary syndrome episodes has been demonstrated to be a clinically and cost-effective model of secondary preventive care. This approach offers several benefits, including improvements in functional capacity, well-being, and quality of life, as well as a reduced rate of hospital readmission and cardiovascular mortality. According to the current guidelines of the European Society of Cardiology, initiating this comprehensive rehabilitation program under the supervision of a multidisciplinary team consisting of cardiologists, physicians, physiotherapists, nurses, psychologists, dieticians, and general practitioners is recommended. During the cardiac rehab program, patients undergo a clinical and paraclinical evaluation to assess their current status and tailor the individualized training prescription. Additionally, patients receive education on managing their risk factors, adhering to medication regimens, receiving dietary advice, and addressing psychosocial aspects. These aspects are particularly crucial in younger patients, as effective management of their condition can lead to a quicker psychosocial integration into socioeconomic life. Furthermore, it is recommended to initiate cardiac rehabilitation in the hospital setting and to continue the program at home, to maintain and further improve their achieved results [12,13].

## 2. Detailed Case Description

A 30-year-old patient was admitted to the Rehabilitation Hospital to participate in a phase II cardiac rehabilitation program following an episode of acute coronary syndrome. He was an ex-smoker (six packs/year, abstinent for one year) and declared no chronic alcohol consumption.

Family history was positive for arterial hypertension and viral C hepatitis on the paternal side, while the mother had a confirmed diagnosis of diabetes mellitus. 

Reviewing the patient’s medical history, the onset of the cardiovascular disease occurred five years ago, when the patient presented to the emergency room with a complaint of left lower limb pain persisting for approximately three days. The clinical examination identified the presence of a painful and tender cord accompanied by erythema along the course of the great saphenous vein. Consequently, the patient was diagnosed with superficial vein thrombosis and was advised to undergo anticoagulant treatment. 

However, three weeks later, the patient visited the hospital, reporting pain and swelling in the left lower limb. Upon further discussion, it was revealed that the patient had chosen to discontinue the prescribed anticoagulant treatment on his own.

The diagnosis of deep vein thrombosis was confirmed through paraclinical investigations (D-Dimers and Doppler ultrasonography), leading to the initiation of treatment with acenocoumarol (2 mg daily after monitoring the INR value). Following the established guidelines for this condition, a risk factor assessment was conducted, which led to the identification of a heterozygous mutation of factor V Leiden, with normal values of protein C and S, negative beta 2 glycoprotein 1 antibodies (IgM and IgG), negative cardiolipin antibodies (IgM and IgG), negative antiphospholipid antibodies (IgM and IgG), negative lupus anticoagulant ratio, antithrombin III within normal range (120.3%), and homocysteine levels within normal range (8.68 µmol/L).

Four years later, the patient was admitted to the emergency department, arriving 2 h after the onset of constrictive chest pain and dyspnea. 

The ECG revealed sinus rhythm, Q wave in aVR, V1-V2, and ST segment elevation in DI and V2-V4 (Figure 1).

Upon admission, laboratory analysis demonstrated elevated levels of myocardial infarction enzymes (CK-MB = 858 U/L, Troponin T = 9954 ng/mL). Subsequently, the patient underwent an emergency coronary angiography, which identified only a thrombotic occlusion in the proximal left anterior descending artery (clear dominant right coronary artery, common trunk, and circumflex artery without lesions), necessitating arterial stenting. Given the presence of thrombophilia, the patient was prescribed dual antiplatelet therapy (clopidogrel 75 mg o.d. and aspirin 75 mg o.d.) in combination with oral anticoagulant treatment (acenocoumarol 2 mg o.d.). At the same time, due to the myocardial damage resulting from the myocardial infarction and the subsequent decrease in left ventricular systolic function (EF = 30%), the patient was prescribed a reduced dosage of a beta-blocker (bisoprolol 2.5 mg b.i.d.) and sacubitril-valsartan (24/26 b.i.d.) in addition to statin therapy (atorvastatin 80 mg o.d.). This fact was necessitated by the patient’s hypotensive state observed during hospitalization. Subsequent adjustments to the dosage were made according to the blood pressure values in the following patient evaluations. 

Five days after the stenting, the patient was admitted to the Cardiovascular Rehabilitation Clinic. At the physical examination, we detected the presence of three xanthelasmas and varicose veins on the lower limbs. The cardiovascular clinical examination revealed BP-95/61 mmHg, HR-81 bpm, rhythmic heart sounds, and a symmetrical peripheral pulse.

Blood tests revealed the following lipid profile: total cholesterol—102 mg/dL, LDL cholesterol—61.1 mg/dL, HDL cholesterol—34.6 mg/dL, triglycerides—137.5 mg/dL.

The ECG showed the presence of a left ventricular aneurysm (Figure 2).

The Ambulatory Blood Pressure Monitoring revealed a non-dipper profile, with a systolic blood pressure value that varied between 84 and 120 mmHg during the day and between 81 and 115 mmHg during the night and a diastolic blood pressure that varied between 41 and 76 mmHg during the day and 45 and 88 mmHg during the night (Figure 3). 

For a complete evaluation, we performed a Holter monitorization in order to find any heart conduction disorder. The mean frequency was 58 bpm, with a minimum of 44 bpm and a maximum of 114 bpm. The patient had 444 ventricular extrasystoles (less than 1% of the total number of monitored beats), mostly isolated; 36 episodes of bigeminy; one trigeminy episode; and rare supraventricular extrasystoles. The patient had no pauses greater than 2.5 s (Figure 4).

The echocardiographic examination identified a dilated left ventricle with hypokinesia observed in the anterolateral wall, apex, and interventricular septum; EF = 30%; moderate ischemic mitral regurgitation; mild tricuspid regurgitation; and no pericardial effusion. Furthermore, the investigation confirmed the apical left ventricle aneurysm (Figure 5).

ESC guidelines recommend a reduction in the LDL-C of more than 50% of the baseline value and <55 mg/dL for subjects at very high cardiovascular risk, such as our patient [14]. Thus, the treatment with atorvastatin 80 mg o.d. was continued. 

Given the administration of acenocoumarol, the International Normalized Ratio (INR) was evaluated. Our patient had a high variability of the INR for minor adjustments in the acenocoumarol treatment.

As part of the comprehensive Cardiac Rehabilitation (CR) program, the patient received multidisciplinary counselling to address and improve their physical, psychological, and social well-being. After thorough assessments, personalized recommendations were provided to educate and support the patient in maintaining optimal cardiovascular fitness, implementing a suitable diet, and adopting a lifestyle that mitigates the risk factors associated with recurrent cardiac events (adopting a Mediterranean diet with a high intake of fibers, more than 200 g of fruits and vegetables per day, minimizing the consumption of processed meat, drinking less than 100 mL alcohol per week, smoking cessation). 

An important aspect to consider within the realm of CR is the observation of a notable benefit concerning the patient’s mental status. During the initial evaluation, the patient exhibited significant major depressive symptoms (23 points), as indicated by the Beck Depression Inventory. However, at the end of the CR program, the patient’s depressive symptoms had notably diminished to a mild level, reflected by a score of 16 points. 

The physical activity within the rehabilitation program must be performed so that the patient reaches a target heart rate equal to 80–85% of the maximum heart rate obtained during the effort test. Since the patient was in the early rehabilitation period, we decided to perform only the 6 min walk test to avoid possible risks that may occur during the cardiopulmonary exercise test. During the 6 min walk test, the patient walked 237 m before the onset of dyspnea.

As a result, a customized aerobic exercise training regimen was implemented, considering the patient’s characteristics. The program was designed to span a two-week period, with five exercise sessions scheduled per week.

Each session started following a 5–10 min warm-up (1) and ended with a 10 min cool-down exercise (3) (Figure 6). Close monitoring of the vital signs was maintained throughout the session to ensure the patient’s safety.

The actual activity lasted 10 min and was carried out at intervals. The patient performed only aerobic training, as the resistance activity was prohibited for his condition. The program was divided into periods of 5 min, in which the patient performed physical activity at 15 W, followed by a period of 5 min in which the patient had an activity of 5 W. In time, if the patient’s condition allows, this type of training can be extended up to 40 min (Figure 7).

Notably, throughout the program, the patient exhibited a remarkable improvement in exercise adherence, successfully completing the prescribed exercise without experiencing dyspnea (Figure 8).

Additionally, notable improvements were observed in the patient’s chronotropic and vasopressor function. These changes indicated that the patient’s cardiovascular system had adapted favorably to the exercise protocol.

After completing the cardiovascular rehabilitation program, an echocardiography was performed to assess the cardiac status. The results revealed a modest improvement in the systolic function of the left ventricle (EF = 44%). Additionally, slight hypokinesia of the interventricular septum and anterolateral wall and apex dyskinesia (aneurysm) were observed. Furthermore, mild regurgitation was noted in the mitral and tricuspid valves (Figure 9).

Regarding the regular assessment of the INR, a suboptimal level control was noted. Given the challenges associated with maintaining stable INR levels, the possibility of utilizing a direct oral anticoagulant, such as apixaban 5 mg b.i.d., was taken into consideration as an alternative therapeutic approach.

## 3. Discussion

The incidence of venous thromboembolism increases with age, primarily due to prolonged immobility and various comorbidities. Conversely, the occurrence of this pathology in young adults is linked to different causes, necessitating a comprehensive assessment of risk factors. This assessment aids in promptly identifying the underlying cause of the condition and implementing preventive measures to minimize the risk of morbidity and mortality. Considering the patient’s young age and history of thromboembolism as well as an acute coronary syndrome, it is important to consider the possibility of atrial fibrillation, which could increase the risk of thromboembolic events. The medical literature has identified several risk factors for atrial fibrillation in younger patients, including hypertension, hypothyroidism [15], participation in endurance sports [16], alcohol consumption, and smoking [17]. Additionally, the presence of cardiac diseases such as cardiomyopathies [18] and channelopathies are also recognized as risk factors in this patient population [19]. However, in our case, clinical and paraclinical evaluations have ruled out the presence of atrial fibrillation. 

Studies reported that patients with factor V Leiden have a higher incidence of atherothrombotic events, highlighting the necessity of individualizing the antithrombotic treatment regimen. This individualization approach involves selecting suitable single or dual antiplatelet agents based on the patient’s underlying phenotype and the timing of the stent implantation. Furthermore, in alignment with the current guidelines, it is recommended to adhere to a 12-month duration of dual antiplatelet therapy following an episode of acute coronary syndrome [20]. However, patients identified as having a low risk for bleeding complications may necessitate an extended duration of this therapeutic regimen. Thus, a multinational registry study revealed that carriers of factor V Leiden diagnosed with venous thromboembolism had 50% lower chances of major bleeding events when these patients underwent oral anticoagulant treatment [21]. To further evaluate the bleeding risk, Mahmoodi et al. were the first to conduct a pooled analysis of three randomized clinical trials involving patients with factor V Leiden and acute coronary syndromes who required long-term antiplatelet therapy. Among 17,623 patients, a notable 5.5% of those with factor V Leiden developed major or minor bleeding events. Therefore, this study reported a reduced bleeding risk associated with factor V Leiden in patients with acute coronary syndrome who received dual antiplatelet therapy [22].

According to current studies, patients diagnosed with deep vein thrombosis and thrombophilia are generally recommended a long-term anticoagulation therapy, with vitamin K antagonists being considered the gold standard [23]. However, in this case, the targeted INR range of 2–3 could not be achieved, and alternative solutions needed to be pursued. The utilization of direct oral anticoagulants remains a topic of controversy, primarily due to the limited availability of comprehensive data supporting their efficacy and safety profiles in this category of patients [24]. Although the 2016 American College of Chest Physicians guidelines acknowledge the potential use of DOACs for the treatment of deep vein thrombosis, there is a lack of specific recommendations regarding the choice of anticoagulant in patients with thrombophilia [25]. 

Therefore, Elisabaie et al. conducted a meta-analysis to evaluate the utilization of DOACs in patients diagnosed with deep vein thrombosis and thrombophilia. The results indicated beneficial effects of DOACs in terms of both treatment efficacy and the secondary prevention of venous thromboembolism in patients with thrombophilia. Furthermore, the data revealed similar rates of VT recurrence, as well as minor and major bleeding risks, between DOACs and VKAs [24]. In another study involving a total of 4866 patients, among whom 446 were diagnosed with thrombophilia, a reduced incidence of clinically relevant non-major bleeding events was observed in patients treated with DOACs compared to those receiving conventional treatment. Also, when assessing the risk of major bleeding and thrombotic events, no significant differences were found between the two therapeutic regimens [26]. Regarding the choice of anticoagulant, a study analyzing patients with thrombophilia who received DOACs stated that rivaroxaban therapy was associated with an incidence rate of 3.72 DVT episodes per 1000 patient-years, while apixaban demonstrated a comparatively lower rate of 1.99 events per 100 patient-years. Additionally, the study revealed a slightly increased risk of bleeding in patients treated with rivaroxaban when compared to those receiving apixaban or dabigatran [27]. 

When considering the aspect of long-term anticoagulation, an essential factor to consider is the appropriate dosing of these medications. A study conducted on a cohort of 209 patients aimed to evaluate the benefits of DOACs for secondary prophylaxis of TEV in patients diagnosed with thrombophilia compared to a group without this underlying pathology. Following a 20-month follow-up period, the study findings revealed no statistically significant differences in safety and efficacy outcomes between full-dose and reduced-dose DOAC regimens [28]. Thus, in this patient’s case, the decision was made to continue the treatment using apixaban at a dosage of 2.5 mg twice daily and double antiaggregant treatment for the first six months after the MI. After this period, the patient will be clinically reevaluated and paraclinical using Doppler ultrasound, echocardiography, and chest radiography. We consider that the patient will need further OAC treatment associated with only one antiaggregant medication. Further guidelines are required for this particular category of patients.

Following an acute coronary syndrome episode, CR programs have demonstrated benefits in reducing the risk of major cardiac events. However, despite these advantages, adherence to such programs remains suboptimal. In the case of this young patient with a history of TEV and acute coronary syndrome, active participation in a comprehensive CR program resulted in significant improvements in cardiac function. As a result, the patient experienced enhanced reintegration into the socioeconomic environment, as indicated by the patient’s reported reduction of dyspnea threshold and increased the exercise tolerance by the completion of this program phase. Furthermore, the patient received personalized recommendations regarding exercise training. These recommendations were based on the evaluation of a new cardio-pulmonary test, which provided valuable information about the patient’s cardiovascular and pulmonary function during exercise.

## 4. Conclusions

Thrombophilia is a genetic disorder characterized by a thrombotic state, which, in the presence of certain trigger factors, can occasionally lead to thrombosis occurrence. Early diagnosis of this condition and a clear explanation of the patient’s risks are crucial, as treatment adherence significantly influences the patient’s overall outcome. Moreover, in the case of this patient with suboptimal INR control, the administration of apixaban has shown favorable effects, warranting further studies to explore its potential benefits.

## Figures and Tables

**Figure 1 reports-06-00039-f001:**
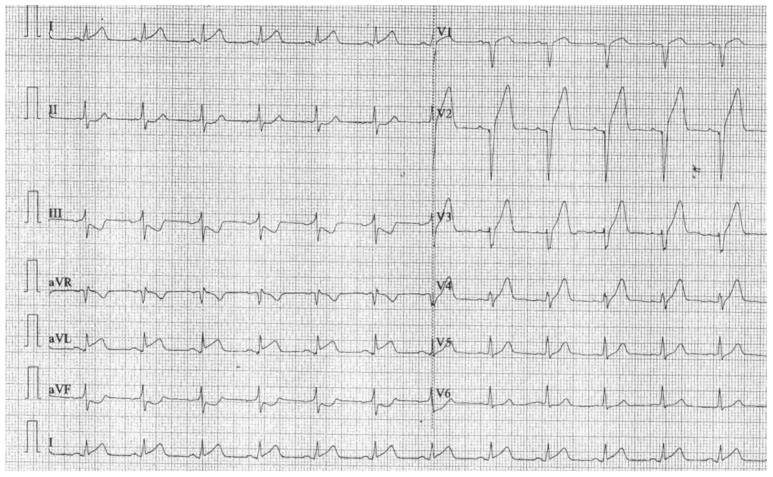
ECG at admission.

**Figure 2 reports-06-00039-f002:**
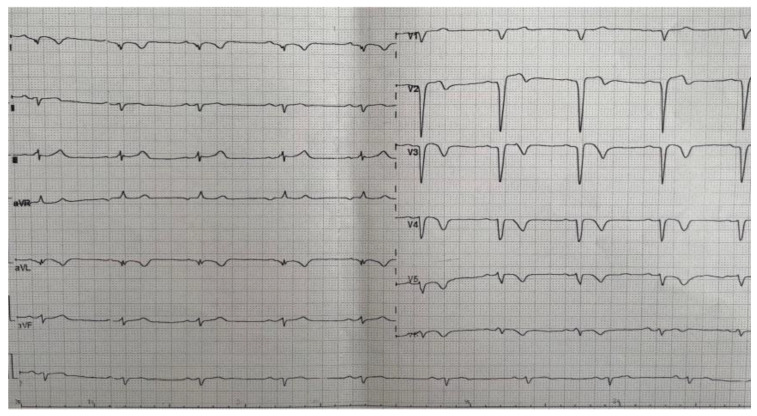
ECG—Left ventricular aneurysm.

**Figure 3 reports-06-00039-f003:**
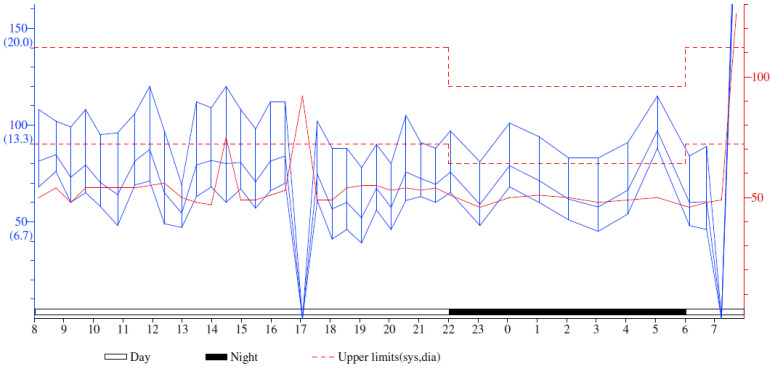
Ambulatory Blood Pressure Monitoring Report. The solid red line represents the pulse throughout the monitorization.

**Figure 4 reports-06-00039-f004:**
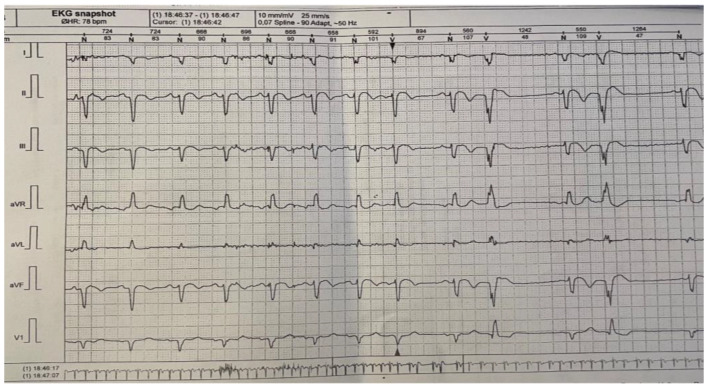
24 h Holter monitoring.

**Figure 5 reports-06-00039-f005:**
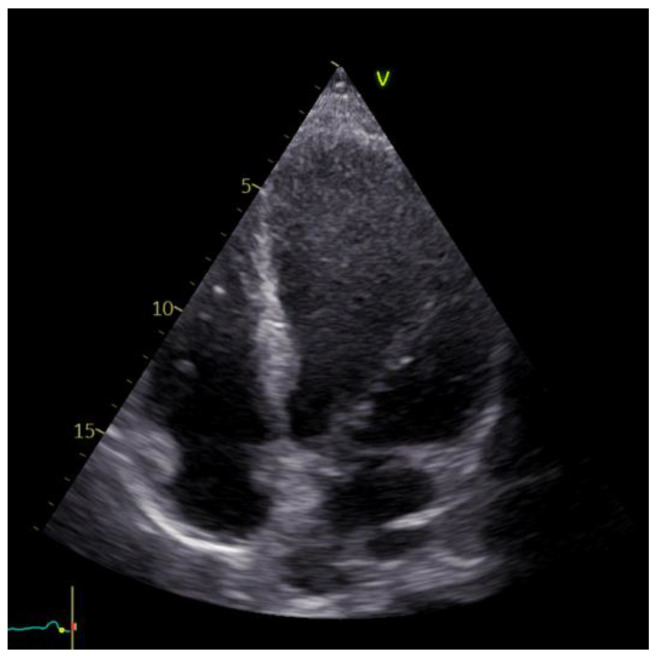
Echocardiographic image of the left ventricle aneurysm.

**Figure 6 reports-06-00039-f006:**
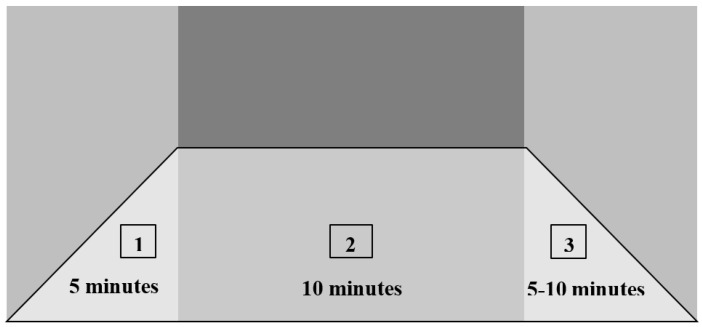
Customized aerobic exercise training: (1) warm-up, (2) activity, (3) cool-down.

**Figure 7 reports-06-00039-f007:**
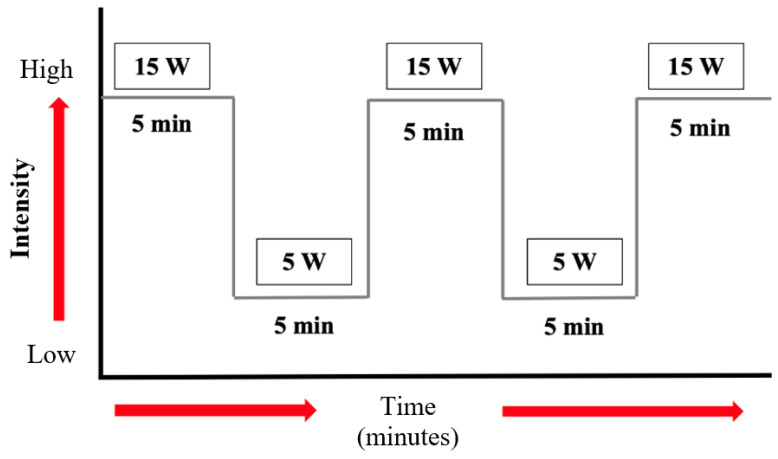
Physical exercise scheme: duration and intensity.

**Figure 8 reports-06-00039-f008:**
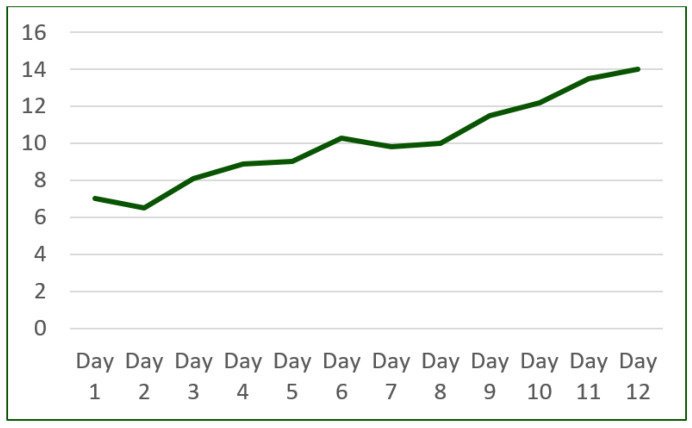
Cardiac rehabilitation program improvement (number of minutes/day).

**Figure 9 reports-06-00039-f009:**
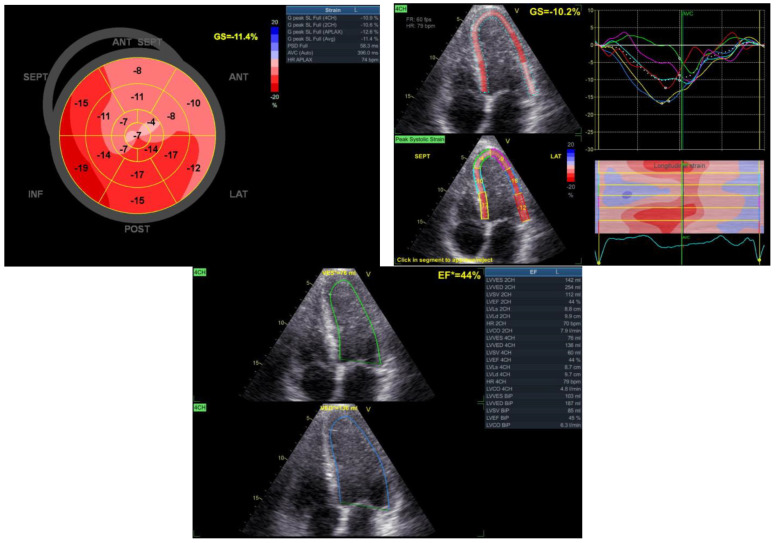
Echocardiographic aspects observed after the rehabilitation program.

## Data Availability

Not applicable.

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
