# Peer review of "Challenges of a Patient with Thromboembolism"

_reports, 2023, doi:10.3390/reports6030039_

Round 1

Reviewer 1 Report

I would congratulate with authors for this very good case report since FV Leiden is an autosomal dominant disease, representing one of the most prevalent genetic causes for hereditary thrombophilia manifested by venous thromboembolism. The case is interesting I have only one consideration: among "Challenges of a patient with thromboembolism" in discussion section, authors should cite other causes of thromboembolism that may also affect young adults (30 y old). In particular, AF in the young may be precipitated by hypertension, hyperthyroidism (doi:10.1001/archinte.164.15.1675), lifestyle factors such as endurance sport (https://doi.org/10.1093/europace/eun289), alcohol consumption, even smoking (doi: 10.1016/j.hrthm.2011.03.038), cardiomyopathies (DOI: 10.1111/jce.14526) and finally channelopathies (doi: 10.1111/jce.13410). Please cite this very important points, including all suggested 5 references.

Author Response

Dear reviewer,

Thank you for all your comments.

We have added in our clinical case at the Discussion section the articles you mentioned. Thank you!

If there are any other changes you consider we should make, please let us know.

Yours sincerely,

All the authors

Reviewer 2 Report

The case is interesting, but the authors` approach is not adequate from my point of view. They have to decide if pathophysiology of Factor V Leiden related thrombosis is to be presented or the accent is on rehabilitation. Detailing the cardiac rehab is not necesarry if the enhanced thrombogenesis is the main topic of the article. There are many missing data related to the case: family history, detailed thrombophylia panel, cholesterol levels, smoking status, angioCT data (other palques on coronaries), exclusion of coronary embolism,  PFO, TOE data.  The timing of PCI is questionable, if we consider the severe LV function deterioration. Or the STEMi occurred on a chronic substrate? The dicussion of long term antithrombotic treatment strategy would be welcome.  English has to be substantially improved.

English has to be substantially improved.

Author Response

Dear reviewer,

Thank you for all your comments.

  1. Our focus is on the rehabilitation of the patients with thrombophilia and MI. We have added in the introduction a paragraph regarding this aspect. Thank you for your remark!
  2. As you mentioned, there were some data missing regarding the family history, thrombophilia panel, cholesterol status, smoking status. Moreover, we have added more details regarding the coronary angiography he has made. We have added them in the article. Thank you very much!
  3. AngioCT will be performed in the next reevaluation. Regarding PFO, this was not observed in the transthoracic echocardiography. Unfortunately, we do not have the possibility to make TOE in our hospital.
  4. The patient was not echocardiographic evaluated in the moment of deep vein thrombosis, this investigation being done for the first time when the MI occurred. Thus, we also assume that the patient had an ischemic chronic substrate, as you mentioned.
  5. We have added in the Discussion section a paragraph where we mentioned the long-term antithrombotic treatment strategy. Thank you!

Hope we have touched all the points you asked us to change.

If there are any other changes you consider we should make, please let us know.

Yours sincerely,

All the authors

Round 2

Reviewer 1 Report

Congratulations to the authors. Manuscript definitely improved!

Reviewer 2 Report

The problems raised were properly treated by the authors.

still has to be improved, ortographic errors have to be corrected